# Molecular Characterization and Selection of Indigenous SARS-CoV-2 Delta Variant for the Development of the First Inactivated SARS-CoV-2 Vaccine of Pakistan

**DOI:** 10.3390/vaccines11030607

**Published:** 2023-03-07

**Authors:** Muhammad Waqar Aziz, Nadia Mukhtar, Aftab Ahamd Anjum, Muhammad Hassan Mushtaq, Muhammad Furqan Shahid, Muzaffar Ali, Muhammad Abu Bakr Shabbir, Muhammad Asad Ali, Muhammad Nawaz, Tahir Yaqub

**Affiliations:** 1Institute of Microbiology, University of Veterinary and Animal Sciences, Lahore 54000, Pakistan; 2Department of Epidemiology, University of Veterinary and Animal Sciences, Lahore 54000, Pakistan; 3Veterinary Research Institute, Lahore 53810, Pakistan

**Keywords:** SARS-CoV-2, rhesus macaque, Vero-E6 cell, vaccine

## Abstract

Vaccines are one of the efficient means available so far for preventing and controlling the infection rate of COVID-19. Several researchers have focused on the whole virus’s (SARS-CoV-2) inactivated vaccines which are economically efficient to produce. In Pakistan, multiple variants of SARS-CoV-2 have been reported since the start of the pandemic in February 2020. Due to the continuous evolution of the virus and economic recessions, the present study was designed to develop an indigenous inactivated SARS-CoV-2 vaccine that might help not only to prevent the COVID-19 in Pakistan, it will also save the country’s economic resources. The SARS-CoV-2 were isolated and characterized using the Vero-E6 cell culture system. The seed selection was carried out using cross-neutralization assay and phylogenetic analysis. The selected isolate of SARS-CoV-2 (hCoV-19/Pakistan/UHSPK3-UVAS268/2021) was inactivated using beta-propiolactone followed by vaccine formulation using Alum adjuvant, keeping the S protein concentration as 5 μg/dose. The vaccine efficacy was evaluated by in vivo immunogenicity testing in laboratory animals and in in vitro microneutralization test. The phylogenetic analysis revealed that all the SARS-CoV-2 isolates reported from Pakistan nested into different clades, representing multiple introductions of the virus into Pakistan. The antisera raised against various isolates from different waves in Pakistan showed a varied level of neutralization titers. However, the antisera produced against a variant (hCoV-19/Pakistan/UHSPK3-UVAS268/2021; fourth wave) efficiently neutralized (1:64–1:512) all the tested SARS-CoV-2 isolates. The inactivated whole virus vaccine of SARS-CoV-2 was safe and it also elicited a protective immune response in rabbits and rhesus macaques on the 35th-day post-vaccination. The activity of neutralizing antibodies of vaccinated animals was found at 1:256–1:1024 at 35 days post-vaccination, indicating the effectiveness of the double-dose regime of the indigenous SARS-CoV-2 vaccine.

## 1. Introduction

The first human infection reported in China in 2019 with the severe acute respiratory syndrome (SARS) was identified as a novel β-coronavirus; however, its zoonotic origin is yet to be defined [1].The index case of SARS-CoV-2 in Pakistan was reported in Karachi on 26 February 2020, followed by the country-wide spread of the disease. More than 6200 people contracted the infection, and 111 deaths were reported during the first seven weeks of the commencement of the epidemic of COVID-19 in Pakistan [2].

The vaccines proved to be the only functional defense against the COVID-19 [3]. Moreover, it is also important to develop an indigenous immunologically efficient vaccine by analyzing the phylogeny and viral pathogenicity of prevalent strains of the virus in a country. Globally, four COVID-19 vaccine categories were produced and tested including whole virus inactivated vaccines, purified protein-based vaccines, mRNA, and vector-based vaccines [4]. Among the available vaccine categories, the inactivated cell culture-based vaccines are easy to develop [5]. Regardless of the type, vaccines produce adaptive (humoral and/or cellular) immune responses against the putative virus/bacterium/fungus which, in turn, prevent propagation of the pathogens in the host. Immune response generated by the vaccines, prior to the viral infection, needs to cease after some time for restoration of the individual’s health. Traditional, inactivated or live attenuated vaccines do not interfere with this balance, while mRNA vaccines that linger for more than four weeks may disturb this balance and could compromise the health of the recipient of the vaccine [6,7]. Moreover, the modern day expression vaccines are not efficient in reducing the risk of disease (<2%) [8]. It is important to note that the activated T-cells against one SARS-CoV-2 variant will efficiently cross react with any other SARS-CoV-2 variant [9]. Thus, the humoral responses should be used as a readout system of the effectivity to activate a meaningful adaptive immune response than thinking to be able to judge an in vivo effectiveness of the vaccine against relevant viral respiratory tract infection. The respiratory tract’s immune defense is chiefly based upon the protection rendered by innate immune system while the adaptive one has very limited access to contain respiratory infections [9,10,11].

Since February 2020, multiple lineages have been identified as the variant of concern (VOC), including B.1.1.7 (Alpha), B.1.351 (Beta), B.1.1.28.1 (Gamma), and B.1.617.2 (Delta). These variants have increased transmissibility to cause infection in the naive population and have potential to evade the body’s immune system. Due to the continuous evolution of the SARS-CoV-2 variants, it would be wise to develop vaccines from the most recent strains [12]. In Pakistan, 58.14% (132.48 million people have been vaccinated to date (covid.gov.pk), a considerable proportion of the population is still unvaccinated. Hence, the risk of re-emergence of SARS-CoV-2 is maintained. Therefore, it is the need of the hour to develop an indigenous SARS-CoV-2 vaccine in Pakistan.

The traditional vaccination approach with a whole inactivated virus is appreciable and feasible for a limited resource country such as Pakistan; therefore, the present study aimed to develop a whole virus vaccine against the indigenously circulating variant. The present study will also provide guidelines for developing countries for the inactivated vaccine production for future epidemics.

## 2. Materials and Methods

### 2.1. Ethical Statement

All the protocols used under the current experimental study were evaluated and approved by the Institutional Ethical Review Committee (DR/459) of the University of Veterinary and Animal Sciences (UVAS), Lahore. The experimental animals were handled according to the local and international ARRIVE guidelines.

### 2.2. Isolation of Indigenous SARS-CoV-2 Variant/s from the Clinical Samples

For the isolation of SARS-CoV-2, nasopharyngeal swabs (n = 780)s were collected from different areas of Punjab Province from March, 2020 to November, 2021 in a viral transport medium (VTM) kit (Copan Diagnostics, Murrieta, CA, USA). The samples were tested using real-time PCR (Thermo Fisher Scientific, Waltham, MA, USA) at the Biosafety Level 3 (BSL-3) facility at the Institute of Microbiology University of Veterinary and Animal Sciences, Lahore. A total of 20 nasopharyngeal swab samples with Ct < 25 were selected and processed for the isolation of SARS-CoV-2.

The Vero-E6 cell line (African monkey green kidney cells (ATCC CRL-1586)) was procured from the Defense Science and Technology Organization (DESTO) Pakistan and cultured in Dulbeco’s Modified Eagle Medium (DMEM) (Sigma–Aldrich Company, Burlington, MA, USA). The medium was supplemented with 10% fetal bovine serum (FBS) (Gibco, Waltham, MA, USA). Penicillin (100 IU/mL), streptomycin (100 µg/mL) and amphotericin B (250 μg/mL) (Biological Industries, Kibbutz Beit-Haemek, Israel) were added to the media to avoid contamination.

The representative samples were diluted and inoculated on a Vero-E6 cell line having 80% confluent monolayer and agitated gently for 1 h at 37 °C for adsorption. The DMEM media was added and kept at 37 °C in the presence of 5% CO_2_. The flask was observed for cytopathic effects (CPEs) daily.

### 2.3. Selection of Vaccine Candidates Based on Phylogeny and Cross Neutralization

Thirteen SARS-CoV-2 isolates were sequenced via next-generation sequencing [13] (The Quadram Institute, Norwich, UK). All sequences were uploaded to NCBI and GISAID databases and accession numbers (MW031799 to MW031803, EPIISL548942-EPIISL548948, EPIISL5063601) were obtained. The sequences of SARS-CoV-2 isolates were compared with publicly available SARS-CoV-2 sequence databases (GISAID and NCBI) for phylogenetic analysis. Large dataset phylogenies of newly sequenced SARS-CoV-2 isolates, and maximum likelihood phylogenetic tree was constructed using FastTree in Geneious v7.1.9 (Biomatters Ltd., Auckland, New Zealand). The dataset were then sub-sampled to 3395 and compared with the 13 sequenced isolates of SARS-CoV-2 of the present study.

For assessing the cross-neutralization, hyper-immune sera were raised for each of the variants in COVID-19 negative-laboratory animals (rabbits) at the BSL-3 animal containment facility [14].

### 2.4. Virus Titration

The tissue culture infective dose 50% (TCID50) of the indigenous SARS-CoV-2 vaccine candidate virus was calculated following the Reed and Muench method [15]. Briefly, the propagated virus was diluted 10-fold in DMEM (Gibco), and 100 μL from each dilution was seeded to each well of a microtiter plate with 80% confluency. The plate was incubated at 37 °C with 5% CO_2_, and cytopathic effects were observed after 72 h. As 100 uL of the virus was used to perform the TCID50, after obtaining the results, the final value was multiplied by 10 to assume the TCID50 per mL of virus.

### 2.5. Production of Inactivated SARS-CoV-2 Vaccine

The SARS-CoV-2 strain hCoV-19/Pakistan/UHSPK3-UVAS268/2021 was propagated with the multiplicity of infection (MOI: 0.01) in serum-free medium and incubated at 37 °C in a CO_2_ incubator. After CPEs (80–90%) were manifested, the virus was harvested by three freeze–thaw cycles and stored at −80 °C until further use.

For inactivation, the harvested SARS-CoV-2 virus was centrifuged at 2000 RCF for 10 min to separate the cellular debris. The supernatant was harvested and used for virus inactivation using β-propiolactone (BPL) (Solarbio^®^ Life Sciences, Beijing, China) with a final concentration of 1:1600 [16]. The mixture was incubated at 4 °C for 10 h with continuous stirring at 100 rpm. The BPL was hydrolyzed by incubating the tubes at 37 °C. Viral inactivation was confirmed on a cell culture system by seven consecutive blind serial passages with a lack of development of CPEs.

The virus was purified by Pierce™ Protein Concentrator PES, 100 K MWC0 (Thermo Fisher Scientific, Waltham, MA, USA). Briefly, the inactivated virus fluid was centrifuged through a 100 K MWC0 filter. The retentate was retained for further processing. For vaccine formulation, different antigen concentrations were adjusted to 5 µg per dose as described previously [17]. The Imject™ Alum (Thermo Fisher Scientific, Waltham, MA, USA) was used as an adjuvant (250 μg per dose) for the immunogenicity studies in rabbits.

### 2.6. Protein Profile of Viral Antigen

The stability of the receptor binding domain of the S-Protein of the virus before and after inactivation was confirmed and quantified using the RayBio^®^ COVID-19 S-Protein (S1RBD) ELISA kit following the manufacturer’s instructions.

### 2.7. Immunogenicity Studies in Mice, Rabbits, and Rhesus Macaques

The immunogenicity of the indigenous inactivated COVID-19 vaccine was evaluated for its capability to elicit neutralizing antibodies in mice, rabbits, and rhesus macaques. A total of 40 male mice (BALB/c) with weight 18–22 g and age of 8 weeks were procured from Animal Reproduction Department, UVAS. The mice were randomly divided into two groups: group I had 30 mice, and group II had ten mice. The mice in group I were immunized through the intramuscular route, and the booster dose was administered at 14 days post-vaccination. While group II was kept as a negative control and administered with an equal volume of sterile phosphate-buffered saline (PBS). Similarly, rabbits (male, n = 20) were divided into two groups viz group I (n = 15, vaccinated) and group II (n = 5, mock). The non-human primates (rhesus macaques: male, n = 6) were also divided into two groups viz group I (n = 5, vaccinated) and group II (n = 1, mock). Group I of rabbits and non-human primates were vaccinated by an intramuscular route at day 0, followed by a booster at 28 days post-vaccination. While group II of rabbits and non-human primates were administered with an equal volume of sterile PBS. Blood samples from experimental animals were collected on days 0, 7, 14, 21, 28, and 35 post priming and sera were separated and stored at −20 °C for immunological investigations [17].

### 2.8. SARS-CoV-2 Nucleoprotein-Based ELISA

The collected serum samples were tested for anti- SARS-CoV-2 antibodies using ID Screen^®^ SARS-CoV-2 Double Antigen Multi-species ELISA kit (ID.vet, Grabels, France) as per manufacturer’s instructions [18]. Briefly, serum samples, positive and negative controls (25 uL each) were pipetted into respective wells of antigen-coated 96-well ELISA plates. The plates were then incubated at 37 °C for 45 min. The wells were washed thrice with a wash buffer and tapped over a paper towel to drain the excess buffer. Afterward, 100 uL of the freshly prepared conjugate was added into each well and incubated for 30 min at room temperature. The wells were again washed as described earlier. After that, 100 uL of HRP substrate was added into each well and incubated for 20 min at room temperature in a dark place. At the final stage, 100 uL of stop solution was added into each well, and the plates were read immediately at 450 nm. All the samples and controls were processed in triplicate to get the most accurate readings.

### 2.9. Micro Neutralization Assay of Mice, Rabbits, and Rhesus Macaques Serum

The collected sera were also checked for neutralizing antibodies against the SARS-CoV-2 virus. Briefly, flat bottom 96-well microtiter plates were seeded with (3 × 10^5^ cells/mL) and incubated until 80–90% confluency was achieved. The serum samples were preheated at 56 °C for 30 min prior to use. Sera were two-fold diluted in a separate microtiter plate. Afterwards, the diluted sera (1:2 to 1:1024) were mixed with SARS-CoV-2 virus (100 TCID_50_) in equal volume and incubated for one hour at 37 °C and transferred to 96-well microtiter plate containing Vero E6 cells. The plate was incubated for 72 h at 37 °C in a CO_2_ incubator and observed daily for CPEs. The highest dilution of each serum sample which inhibited CPEs in 50% of the tested wells was identified as the TCID50 titer of that serum sample [19].

### 2.10. Toxicity Test in Rabbits and Mice

To determine the single-dose toxicity of the developed vaccine, 25 rabbits were procured and randomly divided into three groups (10 rabbits in groups A and B and five rabbits in group C). Groups A and B were immunized with a standard clinical dose (1000 SU/dose) and a high dose (2000 SU/dose) of inactivated COVID-19 vaccine, respectively, via intramuscular (I/M) route, while Group C was inoculated intramuscularly with normal saline. A similar procedure was adopted for 25 BALB/c mice injected with a total volume of 0.3 mL/dose through the I/M route. After each injection, a local tolerance test for 72 h was assessed [17].

The possible toxicity reaction of the target organ was assessed after administration of SARS-CoV-2 vaccine. A total of 20 rabbits with similar bodyweights were randomly divided into two groups (A and B). Groups A and B served as the treatment and control groups, respectively. The rabbits in groups A and B were administered with 0.5 mL/1000 SU/dose and 0.5 mL/dose of normal saline, respectively. The vaccine safety was assessed using intramuscular injections on days 1, 8, and 15 till four weeks post-last-administration. All animals were monitored for clinical signs and mortality throughout the dose toxicity test. Furthermore, all experimental animals were weighed periodically, and feed consumption was recorded. Samples for clinical biochemistry and hematology were taken on days 2, 21, and 28, as described earlier [17]. 

### 2.11. Statistical Analysis

For analysis of the data, GraphPad Prism version 8.4.3 software was used [20]. The statistical significance was assessed using the Kruskal–Wallis between the control and the vaccinated groups.

## 3. Results

### 3.1. Isolation of Indigenous SARS-CoV-2 Variant/s from the Clinical Samples

A total of 20 nasopharyngeal swab samples with Ct < 25 were selected from 780 samples and processed for the isolation of SARS-CoV-2. A total of 13 samples showed the cytopathic effect on the Vero E6 cell line, as shown in Table 1 and Figure 1.

### 3.2. Selection of Vaccine Candidates Based on Phylogeny and Cross Neutralization

A maximum likelihood phylogenetic tree was constructed for different waves separately. Our phylogenetic analysis revealed that multiple variants have been circulating in Pakistan since February 2020. The first wave of the SARS-CoV-2 infection was observed from February to May 2020. The results showed two major clades which were circulating during the first wave of COVID-19 in Pakistan. The second wave was observed from October to December 2020, and all reported sequences nested in a separate clade (Nextstrain clade classification). From May 2021 to June 2021, the Mu variant was predominantly circulating and nested in a separate clade, whereas from June to October 2021, the Delta variant was mainly found circulating and that was nested in separate clade (Figure 2). 

In addition to the genetic characterization of the SARS-CoV-2 isolates, anti-SARS-CoV-2 polyclonal sera were raised against the representative isolates in rabbits and cross neutralization results are shown in Table 2.

The hCoV-19/Pakistan/UHSPK3-UVAS268/2021 (EPIISL5063601) virus was selected for vaccine preparation. The virus was re-confirmed through real-time PCR and subsequently processed for virus inactivation.

### 3.3. TCID50 of the Vaccine Candidate Virus

The tissue culture infective dose 50% (TCID50) of the indigenous SARS-CoV-2 vaccine candidate virus was calculated as 10^7.5^ virus particles.

### 3.4. SARS-CoV-2 Inactivated Vaccine Preparation

Different concentrations of the BPL (1:1000, 1:1300, 1:1600, and 1:2000) were used for the virus inactivation, and the 1:1600 concentration showed the complete inactivation of the virus having no CPEs in the seven blind passages. The virus was purified by Pierce™ Protein Concentrator PES, 100 K MWC0 (Thermo Fisher Scientific, Waltham, MA, USA). After concentrating the inactivated virus, the final concentration was found to be 1074 μg/mL, which was adjusted to 5 μg/dose of the vaccine.

### 3.5. Protein Profile of Viral Antigen

The quantification of the intact RBD of the S protein, immunologically determinant of SARS-CoV-2 virus was 1989.6 pg/mL.

### 3.6. SARS-CoV-2 Nucleoprotein-Based ELISA 

ELISA results showed that the formulation having 5 μg/dose produced anti-SARS-CoV-2 antibodies in all animal models (mice, rabbits, and rhesus macaques).

### 3.7. Anti-SARS-CoV-2 Antibody Response in Animal Models

No adverse effects or clinical signs were observed during the experimentation in all the animal models (mice, rabbits, and rhesus macaques). The collected serum samples were processed for determination of presence of anti-SARS-CoV-2 immunoglobulin (IgG) antibodies. The results revealed that the inactivated SARS-CoV-2 vaccine developed in this study produced detectable IgG antibodies after three weeks. There was an increased level of antibodies IgG after 35 days of vaccination in all three animal models. The serum samples collected from all animals of the negative control group had no anti-SARS-CoV-2 IgG antibody, as shown in Figure 3. 

### 3.8. Micro Neutralization Assay

The results of microneutralization assay revealed that the serum samples of all vaccinated animals inhibited the development of CPEs by SARS-CoV-2 in Vero-E6 cells. The level of neutralizing antibodies on day 28 was 1:64 to 1:256, while on day 35, it was 1:256 to 1:1024 in all vaccinated animals. In comparison, serum samples from all the animals in control groups never showed any ability to neutralize the virus as found by the observation of CPEs.

### 3.9. Toxicity Test in Rabbits and Mice

The absence of clinical aberrant response and death was used as a measure of toxicity in the mice and rabbits. Our findings revealed no adverse effect on the animals’ bodyweight and feed intake. The primary organs and tissues of animals in each group were free of any gross abnormalities. Even after the administration of the double dose of vaccine, the results showed no abnormal drug administration effects, indicating that the maximum tolerated dose (MTD) was equivalent to or higher than 2000 SU/dose.

In order to determine the toxicity of SARS-CoV-2 vaccine after three administrations, no adverse reaction was found in clinical observations in rabbits of groups (A and B).

### 3.10. Post-Vaccination Blood Profile of Non-Human Primates

Results of the blood profile of non-human primates (rhesus macaques) on different days after the immunization revealed no difference in blood parameters among vaccinated and control animals as shown in Table 3 and Table 4.

## 4. Discussion

Globally, several vaccines have been developed and granted permission for emergency use to combat the spread of SARS-CoV-2, including viral vectored vaccines (ChAdOx1 nCoV-19 and Ad-26.COV.2.S), the mRNA vaccine (mRNA-1273), the DNA vaccine (INO-4800), and inactivated vaccines (PiCoVacc and BBIBPCorV). However, clinical trials (phase III clinical trials) for the (mRNA-1273) and BNT162b2 (BioNTech/Pfizer) have been completed so far [17]. Due to rigorous cold-chain requirements of mRNA-based vaccines and their high prices, the worldwide access to these vaccines is limited, particularly in lower-income and lower-middle income countries such as Pakistan [21]. Therefore, more SARS-CoV-2 vaccines are required to fulfill the worldwide demand and to achieve herd immunity. Most vaccines developed are based on the SARS-CoV-2 spike protein only [22]. The advantage of inactivated whole virus vaccine over the other vaccines is that the immune system response to the inactivated whole virus vaccine would target all of the virus’s proteins, not only the spike protein. Moreover, the whole inactivated virus vaccine provides complex antigenicity and better adaptive or humoral immune response for better cross-reactivity [23]. Traditional vaccines such as the inactivated vaccines have been better explored and the risks associated with their use are better defined, whereas the risk/benefit ratio of mRNA vaccines are not well defined. Some of the recent publications have indicated that the risk of getting sick or having unwanted effects after using the mRNA-based vaccines is higher [24]. It is also supported from the previous publications that expression of spike protein in mRNA vaccine is uncontrollable and possibly disturbs the adaptive immune response due to the presence of artificially stabilized mRNA of the vaccine in the germinal center of secondary lymph organs [6,7].

Previously reported literature has mentioned that the vaccines developed using indigenous strains of SARS-CoV-2 are safe and effective in producing protective antibodies [25]. The current study demonstrated that indigenous inactivated whole virion (SARS-CoV-2) vaccine efficiently elicit immune responses in three animal models (mice, rabbits, and rhesus macaques) and produce neutralizing antibodies against the SARS-CoV-2 virus. Here, our results showed that the efficient immune response among experimental animals, including mice, rabbits, and rhesus macaques, was observed after a two-dose immunization regimen of the indigenous inactivated COVID-19 vaccine.

The selection of indigenous SARS-CoV-2 was carried out by phylogenetic analysis. The results revealed the circulation of several variants, which nested well in separate clades during the same period. This shows the diversification of circulating SARS-CoV-2, which might indicate the direct introduction of SARS-CoV-2 into Pakistan through international trade/travelers or globalization. The phylogenetic analysis of this study is in accordance with the previous reports which have reported the circulation of seven clades of SARS-CoV-2 in Pakistan during the first three waves [26]. These different variants of SARS-CoV-2 originated mainly due to the mutations in the spike protein, which ultimately leads to concerns of reduced vaccine efficacy [27,28].

The vaccine was formulated with aluminum hydroxide, which is considered as the safest and most widely used adjuvant [29,30]. In present study, the safety evaluation of the vaccine formulated with aluminum hydroxide showed no toxic effect in all the experimental animals (mice, rabbits, and rhesus macaques). In general, aluminum hydroxide adjuvant formulations produced significant titers of neutralizing antibodies (NAbs). Neutralizing antibodies are essential in protecting against SARS-CoV-2 infection and have been used to assess vaccination effectiveness as an immunological correlate of protection [31]. In preclinical trials, inactivated SARS-CoV-2 vaccine candidates were demonstrated to generate substantial levels of antigen binding and NAb titers [32,33].

Preclinical studies of the vaccine are critical to evaluate the immunogenicity of the vaccine. A mouse model and non-human primates have been widely used in different studies to evaluate the immunogenicity and effectiveness of the SARS-CoV-2 vaccines [34,35,36]. The immunogenicity of indigenous inactivated whole virion SARS-CoV-2 vaccine in all three animal models (mice, rabbit, monkey) showed a detectable level of antibodies IgG at 28 days of vaccination. In comparison, many studies showed a detectable level of antibodies at 21 days of vaccination [34,36,37,38,39]. This difference may be due to using different ELISA kits for the SARS-CoV-2 nucleoprotein IgG in the tested animal models. In other SARS-CoV-2 inactivated vaccine candidates such as PiCoVacc and BBIBP-CorV, studied in a non-human primate model, the NAb were observed from the first and second week, respectively, with a period of detection lasting for five weeks [36,40].

For the viral vaccine evaluation, neutralization ability is an important indicator. In the study, we also presented the in vitro evaluation of the vaccine using a microneutralization assay. For the subject matter, serum samples of vaccinated animals were used to neutralize SARS-CoV-2 and all the samples neutralized the virus and did not present CPEs on Vero-E6 cells. The antibody titer of all experimental animals vaccinated with SARS-CoV-2 inactivated vaccine was found to be 1:64 to 1:256 at the 28th day of vaccination while this titer was increased at 35th day of vaccination (1:256 to 1:1024). A previous study also reported that the booster immunization produces a high titer of neutralizing antibodies against SARS-CoV in mice (BALB/c) [41]. Without an effective antiviral drug against SARS-CoV-2, vaccines with good potency and safety are needed to achieve herd immunity [42].

The present study also had some limitations including the unavailability of data on T helper cells, which can also contribute towards the humoral response. To overcome this limitation, we determined the antibodies in response to the SARS-CoV-2 vaccine in three different animal models. Other limitations included the use of a small number of rhesus macaques for in vivo studies, unavailability of data on cross-neutralizing ability of this NAb with other related viruses such as SARS-CoV. However, previous studies have reported the cross-neutralizing ability of two SARS-CoV NAb to neutralize SARS-CoV-2 [42]. 

In conclusion, this study presents molecular characterization and selection of an indigenous SARS-CoV-2 variant to develop whole virion-based inactivated vaccine against SARS-CoV-2 in Pakistan. The study also reported the preclinical immunogenicity evaluation of this SARS-CoV-2 vaccine formulated in aluminum hydroxide adjuvant in three animal models including (mice, rabbits, and rhesus macaques). The present study revealed that inactivated SARS-CoV-2 vaccine with a two-dose regimen can elicit a robust immune response and may protect the animals challenged with SARS-CoV-2 infection, which requires further investigations. Developing indigenous SARS-CoV-2 vaccines may also save the budget allocated to import the SARS-CoV-2 vaccine in Pakistan. The present study will also provide a platform for the development of such viral vaccines in the country in future. 

## Figures and Tables

**Figure 1 vaccines-11-00607-f001:**
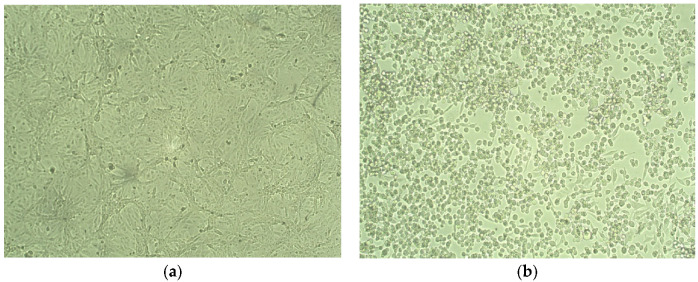
Isolation of SARS-CoV-2. (**a**) Normal Vero Cells (**b**) Cytopathic effect of SARS-CoV-2 after 72 h of infection on vero cells showing plaques and disrupted monolayer.

**Figure 2 vaccines-11-00607-f002:**
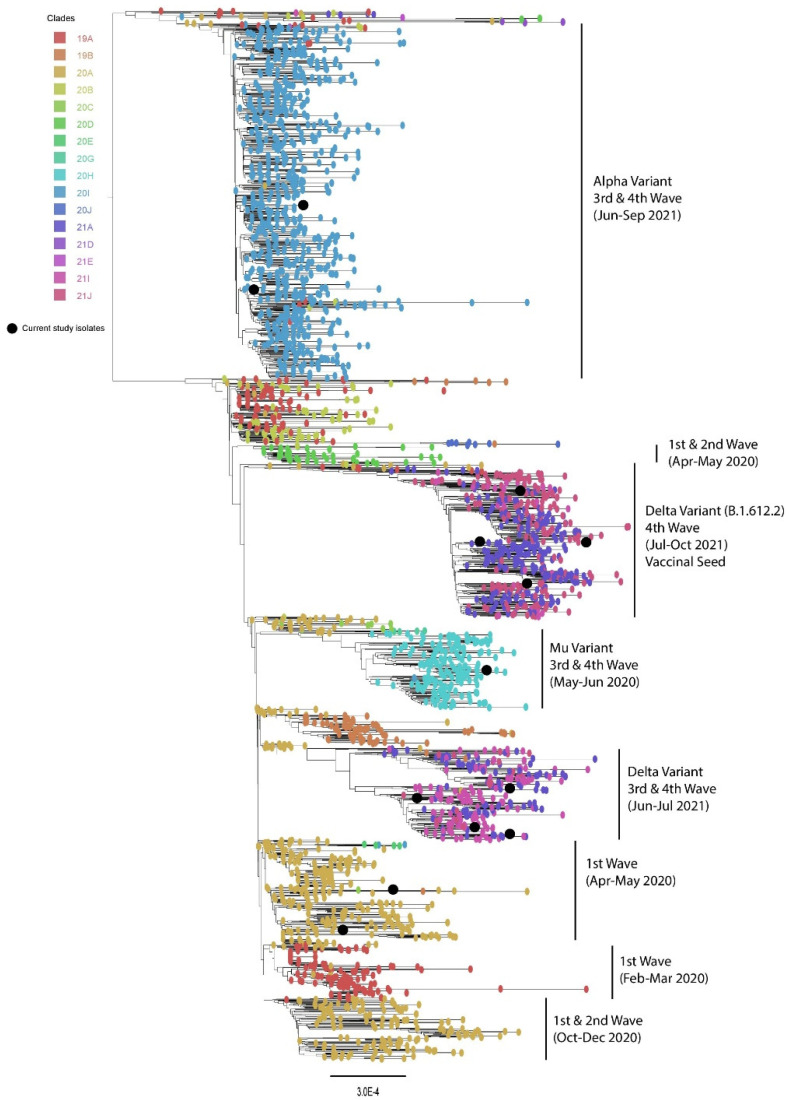
Phylogenetic analysis of circulating SARS-CoV-2 virus strains in Pakistan (black balls) during the 1st–4th wave.

**Figure 3 vaccines-11-00607-f003:**
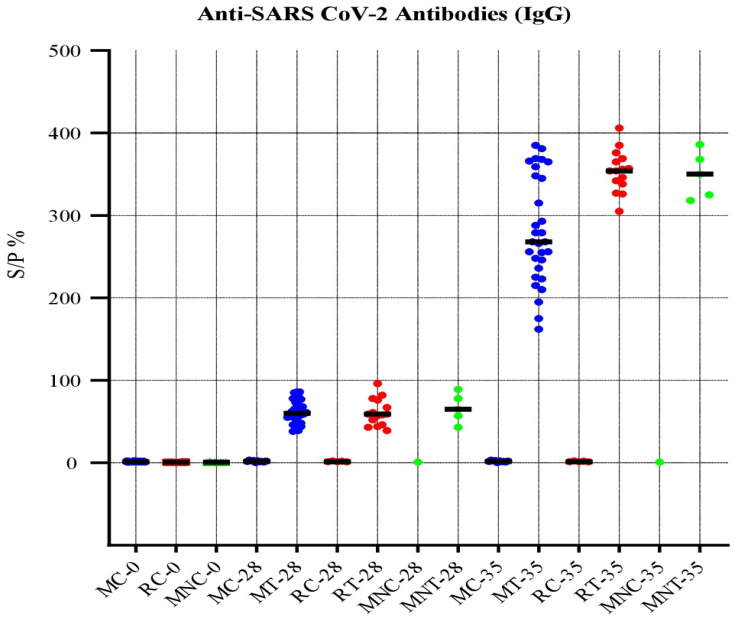
Anti-SARS-CoV-2 antibody response of serum samples.

**Table 1 vaccines-11-00607-t001:** Details of the clinical samples used for virus adaptation.

S.N	Sample ID	Accession Number	Vaccine History	CT
1	7442	MW031799	Sinopharm (double dose)	18.82
2	7501	MW031800	Sinopharm (double dose)	19.6
3	7542	MW031801	Sinopharm (double dose)	22.0
4	7549	MW031802	Sinopharm (double dose)	20.1
5	7305	MW031803	Sinopharm (double dose)	13.37
6	8523	-	Sinopharm (double dose)	15.5
7	8498	-	Sinopharm (double dose)	18.5
8	8494	EPIISL548942	Sinopharm (double dose)	19.5
9	7838	EPIISL548943	Sinopharm (double dose)	18.6
10	7870	EPIISL548944	Sinopharm (double dose)	17.4
11	7869	EPIISL548945	Sinpoharm (double dose)	19.6
12	7878	EPIISL548946	Sinopharm (double dose)	17.8
13	7855	EPIISL548947	AstraZenica (1st dose)	15.2
14	7846	EPIISL548948	Sinovac(1st dose)	14.9
15	8643	-	Sinopharm (double dose)	17.6
16	8649	EPIISL5063601	Sinovav (double dose)	19.9
17	8653	-	Sinovac (double dose)	20.5
18	8842	-	Sinovac (double dose)	19.8
19	8857	-	Sinopharm (double dose)	20.4
20	8910	-	Sinovac (double dose)	18.3

**Table 2 vaccines-11-00607-t002:** Cross-neutralization of different serum samples with viruses.

	EPIISL548942	EPIISL548945	EPIISL548948	MW031799	MW031801	MW031803	EPIISL5063601
EPIISL548942 serum	1:16	1:8	1:4	1:64	1:32	1:32	1:16
EPIISL548945 serum	1:32	1:64	1:128	1:64	1:64	1:32	1:128
EPIISL548948 serum	1:32	1:128	1:128	1:64	1:512	1:128	1:64
MW031799 serum	1:16	1:64	1:128	1:8	1:64	1:32	1:16
MW031801 serum	1:4	1:32	1:16	1:256	1:128	1:64	1:64
MW031803 serum	1:32	1:8	1:16	1:128	1:8	1:32	1:4
EPIISL5063601 serum	1:128	1:256	1:512	1:64	1:128	1:64	1:512

MW031799 to MW031803, EPIISL548942-EPIISL548948, and EPIISL 5063601 are the accession numbers of the uploaded complete sequences of SARS-CoV-2 from Pakistan.

**Table 3 vaccines-11-00607-t003:** Blood profile (post COVID-19 vaccine) of non-human primates (rhesus macaques).

Factor	Monkey 1			Monkey 2			Monkey 3			Monkey 4 Control			Monkey 5		
	0 Day	7 DPV	21 DPV	0 Day	7 DPV	21 DPV	0 Day	7 DPV	21 DPV	0 Day	7 DPV	21 DPV	0 Day	7 DPV	21 DPV
WBC	5.4	10.2	13.16	5.3	4.56	6.06	6.67	7.27	8.21	11.06	12.92	13.2	8.2	11.4	12.73
Lym.	28.6	24.3	22.7	30.1	43	21.2	21.8	39.7	22.4	25	24.3	25.6	9.0	11.9	12
Mon.	5.2	5.1	5.3	5.6	4.6	4.4	4.5	2.3	2.4	3.4	3.0	3.6	4.1	3.4	2.2
Gra.	64.5	69.2	72	64.3	52.4	74.4	73.7	58	75.2	71.6	67.8	72.4	65	78	85.8
RBC	4.65	4.56	5.44	4.51	4.22	4.49	3.97	4.94	5.02	4.56	4.61	5.2	3.85	4.3	4.96
MCV	82.3	78.6	70	81.1	80.3	80.6	64.4	65	64	63.9	66.2	65.3	58	67.9	72.8
Hct	35.6	39.6	38	36.5	33.8	36.1	25.5	32.1	32.1	29.1	30.5	31.9	32.1	38.9	36.1
MCH	23	22.8	22	25.7	27	25.1	22.4	20.8	20.3	19.7	19.7	21.0	18.2	22.5	23.7
MCHC	32.2	33	31.5	31.7	33.7	31.3	34.9	32	31.7	30.9	29.8	33	35.1	36.3	32.6
RDW	10.5	10.9	9.6	10.5	9.6	10.1	10	11	10.6	13.6	14	15.8	10.2	12.6	10.6
Hb	12.3	12.0	12	11.6	11.4	11.3	8.9	10.3	10.2	9	9.1	8.9	8.0	12.9	11.8
THR	203	215	199	234	209	292	286	228	266	256	268	292	366	392	297
MPV	8.3	8.8	8.7	8.2	9.2	8.7	8.1	8.8	8.4	7.5	8.7	7.1	8.3	8.8	9.1
Pct	0.18	0.18	0.17	0.19	0.19	0.25	0.23	0.2	0.22	0.42	0.32	0.38	0.29	0.31	0.27
PDW	8.2	8.1	8.3	8.8	10.1	8.5	8.1	8.3	8.8	7.1	7.5	7.9	7.8	6.9	6.6

**Table 4 vaccines-11-00607-t004:** LFT’s and RFT’s results (post COVID-19 vaccine) of non-human primates (rhesus macaques).

Parameter	M1	M2	M3	M4	M5
**LFT’s**					
Bilirubin (Total)	0.9	2	0.2	1	0.7
ALT (SGPT)	24	25	29	22	34
AST (SGOT)	50	42	54	35	50
Alkaline Phosphatase	339	804	666	265	235
GGT	55	53	65	50	63
Total protein	9	7.3	9.2	8.1	6.2
Albumin	3.9	4.1	3	2.5	4.1
Globulins	5.1	3.2	6.2	5.6	2.1
A/G ratio	0.76	1.2	0.48	0.44	1.9
**RFT’s**					
Creatinine	0.8	0.9	0.6	0.6	0.5
Urea	47	68	40	40	39

## Data Availability

All the data are available.

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
