# Peer review of "Molecular Characterization and Selection of Indigenous SARS-CoV-2 Delta Variant for the Development of the First Inactivated SARS-CoV-2 Vaccine of Pakistan"

_vaccines, 2023, doi:10.3390/vaccines11030607_

Round 1

Reviewer 1 Report

This is a very interesting article regarding the vaccine that will be developed against the indigenously circulating virus variant and will provide the guidelines for the vaccine production for future variants.

It is well written. 

Please include more recent published articles in the discussion.

Please include also the limitations of the present study.

Author Response

Reviewer: 1

Sr. No.

 Comment

Reply

1

Please include more recent published articles in the discussion.

Thank you for pointing out this missing part. The discussion section has been revised at page 11, 12 and recent published articles are included in the revised manuscript.

2

Please include also the limitations of the present study.

Thank you for pointing out this missing part. Limitations have been added in the second last paragraph in the discussion at line 380-393 (Page #12) in the revised manuscript.

Dear reviewer,

Thank you for your valuable suggestions. If there is something that needs further to improve, please guide/suggest us and we will be happy to further revise and clarify the text accordingly.

Reviewer 2 Report

This paper is very poorly written, much too often in pigeon english! Some sentences miss their verb; many are meaningless!  Others have two verbs that differ (for example, "group II was kept... and were administered" ) !!....         Fig 1 and 2 are impossible to read and understand! Table 1 is a long history of virus strains that have received the names  and genders (!!) of their donor (!) while being qualified of "adapted" or not: but no explanation of adaptation is being given! Table 4 is unexplainable and brings no interesting data.

Author Response

Reviewer 2

Sr. No.

 Comment

Reply

1

Poorly Written, Improve the English.

Some sentences miss their verb; many are meaningless!  Others have two verbs that differ (for example, "group II was kept... and were administered" )

Thank you for pointing this error. English grammar has been improved in the revised manuscript by native English speaker.

2

  Fig 1 and 2 are impossible to read and understand!

Thank you for pointing this mistake. High resolution Fig 1 and 2 are added in the revised manuscript.

3

Table 1 is a long history of virus strains that have received the names and genders (!!) of their donor (!) while being qualified of "adapted" or not: but no explanation of adaptation is being given!

Thank you for indicating this mistake. Name and gender details have been excluded in the revised manuscript.

For adaptation, the lines have been rephrased. Please see line 212,213 in the revised manuscript.

4

Table 4 is unexplainable and brings no interesting data.

Thank you for pointing this error. We also found table 3 and 4 are unnecessary, therefore have been removed in the revised manuscript.

Dear reviewer,

Thank you for your Valuable suggestions. If there is something that needs further to improve, please guide/suggest us and we will be happy to further revise and clarify the text accordingly.

Reviewer 3 Report

This study lacks innovation and is not attractive to readers.

Author Response

Reviewer 3

Sr. No.

 Comment

Reply

1

This study lacks innovation and is not attractive to readers.

Dear Reviewer, Thank you for your valuable input. We want to mention that all the previously available vaccines are developed either by using Wuhan reference sequence of SARS CoV-2 (Subunit/Nucleic acid vaccines) or virus isolate of first wave for whole virus vaccine. While the vaccine presented in the present study was developed against Delta variant of SARS CoV-2. This delta variant was having the highest transmission and death rate compared to other variants. This study will provide platform for the development of vaccine in the country to combat the future variants as well.

Secondly, for the developing countries, where no vaccine production facilities are available, development of inactivated vaccine is easy and the present study will provide the guideline for the robust development of inactivated indigenous vaccines.

Please reconsider the article accordingly.

Dear reviewer,

Thank you for your Valuable suggestions. If there is something that needs further to improve, please guide/suggest us and we will be happy to further revise and clarify the text accordingly.

Reviewer 4 Report

The authors have collected extensive data about serum neutralization antibodies from three species including rhesus macaques of a specific inactivated whole SARS-COV-2 vaccine. However, the presentations of these datasets need major revisions. The introduction, discussion and the conclusions need to be rewritten to incorporate following aspects:

The present manuscript is also driven predominantly for publication by following “the major popular narrative”. In so doing, it is a disservice to scientific progress and medicine in general but also to the whole society, e.g. the numbers showcased by the WHO and it is incomprehensible how many  scientists and reviewers are a megaphone for these numbers and this seems to be the case also here in this manuscript. These numbers have nothing to do with serious science and should be purged from any scientific publication: e.g. none of the mucosal based test assays, the base for these WHO numbers, were tested by a ROC analysis as to whether the specificity and sensitivity of the tests were sufficient to use them in population studies. Additionally, based on the WHO own instructions, none of the epidemiologists could understand how many people  were real cases with symptoms and how many died with or from SARS-COV-2.

Moreover, even in the worst case scenario of SARS-COV-2 pathogenicity, the virus caused about 25-100 times less deaths compared to the flu virus, H1N1 virus which caused the Spanish flu in 1918/19. This is additionally supported by the controlled SARS-COV-2 human infection trial published in Nature Medicine in March 2022 (https://www.nature.com/articles/s41591-022-01780-9) where healthy individuals without any evidence of prior exposure to SARS-COV-2 had only symptoms of a normal cold in this infection trial.

This a cautionary note to the authors and should be objectively included in the introduction: neutralization antibodies are not the ultimate judgement about the functionality of the adaptive immune response as the adaptive immune system consists of both the humoral and the cellular immune response and the humoral immune response is also dependent upon the T helper cell function. Additionally, a once triggered T-cell immune response by any SARS-COV-2 variant responds to any of the VOCs by cross-reactivity (https://pubmed.ncbi.nlm.nih.gov/34531555/).

As previously indicated, I miss a critical-objective view by the authors pondering e.g. comparing existing vaccines and their weaknesses/advantages with the vaccine strategy presented here: this could include e.g. whole virus immune complexity and the specific diversification for Pakistan virus variants versus a simple spike protein antigenicity, the uncontrollable expression and distribution through the whole body of spike protein and the mRNA (the artificial stabilized mRNA found also in germinal centers) seen with the new expression vaccines (https://www.ncbi.nlm.nih.gov/pmc/articles/PMC8786601/; https://pubmed.ncbi.nlm.nih.gov/35202565/ ).This is a ticking time-bomb of toxicological side effects by the new expression vaccines (https://www.sciencedirect.com/science/article/pii/S0264410X22010283?via%3Dihub).

Specific uncertainties of the results in present manuscript: Please indicate what the abbreviations HRP, CPE, ..SU/dose, S/P%, in Figure 3, in Table 4 and in Table 5 mean.

It is not at all clear what experiments were done with what animals: rabbits, mice or rhesus macaques. Additionally, after how many immunizations were the serums taken to test their effects in the in vitro assays. This definitively needs to be improved before I can even make a judgement about the data.

-“The datasets were then sub-sampled to 3395 best representing the diversifica-tion of the SARS CoV-2 virus circulating in Pakistan.“

The authors description of Figure 2 is confusing and needs to be better described. Also, did the authors compare the 13 sequenced strains of their own with another 3395 sequences, and of which database?

-“TCID50 of the vaccine candidate Virus 

The tissue culture infective dose 50% (TCID50) of the indigenous SARS CoV-2 vaccine candidate virus was calculated as 107.5 TCID50/ml following Reed and Muench method (1938).“

Are these 107.5 virus particles or 107.5 TCID50/ml dosage?

Author Response

Reviewer 4

Sr. No.

 Comment

Reply

1

The introduction, discussion and the conclusions need to be rewritten to incorporate following aspects.

Thank you for your valuable input. As a student of science, we appreciate your concerns not only for this study but also those that are published previously by others originating from different geographical regions.  However, we followed previously published guidelines/procedures by scientific community/WHO available to-date to conduct the research study presented herein. The obtained study outcomes are presented and discussed accordingly. Also, per your suggestion, we have significantly improved the introduction, discussion and conclusion section of the manuscript.

2

Specific uncertainties of the results in present manuscript: Please indicate what the abbreviations HRP, CPE, SU/dose, S/P%, in Figure 3, in Table 4 and in Table 5 mean.

Thank you for pointing this lacking aspect. An abbreviation list has been inserted at the end of the revised manuscript. (Page 13,14)

3

It is not at all clear what experiments were done with what animals: rabbits, mice or rhesus macaques. Additionally, after how many immunizations were the serums taken to test their effects in the in vitro assays. This definitively needs to be improved before I can even make a judgement about the data.

Thank you for pointing this mistake. Now in each section we have mentioned clearly that experiments were done with what animals: rabbits.

4

The authors description of Figure 2 is confusing and needs to be better described. Also, did the authors compare the 13 sequenced strains of their own with another 3395 sequences, and of which database?

Thank you for pointing this mistake. High resolution Fig 2 is added in the revised manuscript. Our 13 study samples were compared with these sequences data base and now in the Fig 2, these sequences are mentioned with black balls for clarity.

5

TCID50 of the vaccine candidate Virus 

The tissue culture infective dose 50% (TCID50) of the indigenous SARS CoV-2 vaccine candidate virus was calculated as 107.5 TCID50/ml following Reed and Muench method (1938).“

Are these 107.5 virus particles or 107.5 TCID50/ml dosage?

The tissue culture infective dose 50% (TCID50) of the indigenous SARS CoV-2 vaccine candidate virus was 107.5 virus particles. Per dose of the vaccine was adjusted according to concentration of Spike protein (5ug/dose).

Dear reviewer,

Thank you for your Valuable suggestions. If there is something that needs further to improve, please guide/suggest us and we will be happy to further revise and clarify the text accordingly.

Reviewer 5 Report

The authors aimed to determine the development of first inactivated SARS CoV-2 vaccine of Pakistan. Their results demonstrates that inactivated SARS CoV-2 vaccine with two-dose regimen elicited a strong immune response in pre-clinical trials. Moreover, the development of indigenous COVID-19 vaccine could save a huge budget amount allocated to import COVID-19 vaccine in Pakistan.

The study covers some issues that have been overlooked in other similar topics. The structure of the manuscript appears adequate and well divided in the sections. Moreover, the study is easy to follow, but few issues should be improved. Some of the comments that would improve the overall quality of the study are:

a. Authors must pay attention to the technical terms acronyms they used in the text.

b. Please better stated the aim of the study in the abstract and introduction section.

c. The authors stead (page 13) "In the absence of an effective antiviral drug against SARS-CoV-2, vaccines with good  potency and safety will be needed to effectively establish immunity in population ".

This reviewer suggest, to add the a reference in support (please see doi: 10.3390/ijerph191710712)

d. Conclusion Section: This paragraph required a general revision to improve the same and to add some "take-home message".

Author Response

Reviewer 5

Sr. No.

 Comment

Reply

1

Authors must pay attention to the technical terms acronyms they used in the text.

Thank you for pointing this lacking aspect. An abbreviation list has been inserted at the end of the revised manuscript. (Page 13,14)

2

Please better stated the aim of the study in the abstract and introduction section.

The aim of the study are stated in the revised manuscript. At line 14-16 in the abstract and line # 70-75 in the introduction.

3

The authors stead (page 13) "In the absence of an effective antiviral drug against SARS-CoV-2, vaccines with good potency and safety will be needed to effectively establish immunity in population ".

This reviewer suggest, to add the a reference in support (please see doi: 10.3390/ijerph191710712)

Thank you for the valuable input. Reference has been added at line 379, Page 12 in the revised manuscript.

4

Conclusion Section: This paragraph required a general revision to improve the same and to add some "take-home message".

Thank you for your valuable input. The conclusions has been revised at the page 12, line 394-404.

Dear reviewer,

Thank you for your Valuable suggestions. If there is something that needs further to improve, please guide/suggest us and we will be happy to further revise and clarify the text accordingly.

Reviewer 6 Report

This is a great work which may be published.  However, few comments must be addressed:

To add a limitation topic in the discussion and improve the quality of figures.

Author Response

Reviewer 6

Sr. No.

Comment

Reply

1

To add a limitation topic in the discussion and improve the quality of figures.

Thank you for pointing this lacking aspect. Limitations have been added in the second last paragraph in the discussion section at line 380-393 (Page #12) in the revised manuscript.

High resolution Figures have been added in the revised manuscript

Dear reviewer,

Thank you for your Valuable suggestions. If there is something that needs further to improve, please guide/suggest us and we will be happy to further revise and clarify the text accordingly.

Round 2

Reviewer 2 Report

This paper has been written in pigeon english. Several sentences lack a verb (l.194; l.240-242; l.301-302), others are just meaningless (l. 109; l. 114-115; l. 194; l.301-303; l. 347; l. 375) still others are full of repetitions, such as 'However"...(l. 391-392). 

Author Response

Reviewer 2

Sr. No.

 Comment

Reply

1

This paper has been written in pigeon english.

Thank you for pointing this error. English grammar has been improved in the revised manuscript by native English speaker.

2

Several sentences lack a verb (l.194; l.240-242; l.301-302),

The mentioned lines are rephrased or omitted, where there is needed, in the revised manuscript.

3

 Others are just meaningless (l. 109; l. 114-115; l. 194; l.301-303; l. 347; l. 375)

Most of the mentioned lines are omitted while some are rephrased according to the need.

4

still others are full of repetitions, such as 'However"...(l. 391-392). 

The repetition words are removed in the revised manuscript.

Dear Reviewer,

We tried our best to revise the manuscript for the improvement of English grammar for the readers.

We really appreciate your valuable suggestions to improve our manuscript. If there is something that needs further to improve, please guide/suggest us and we will be happy to further revise and clarify the text accordingly.

Reviewer 3 Report

Agree with the author's explanation, and recommend accepting in present form. 

Author Response

Reviewer 3

Sr. No.

 Comment

Reply

1

Agree with the author's explanation, and recommend accepting in present form. 

Thank you so much for accepting the manuscript.

Dear reviewer,

We thank you for giving your valuable time for reading this manuscript keenly. We also appreciate that you understand our explanation and recommend the manuscript.

Reviewer 4 Report

I was glad to be of help to improve the presentation of the Results. As requested in your recent rebuttal, I will try to help with your introduction, discussion and conclusion.

Unfortunately, the WHO has overreaching power as an health as well as a political organization. This would not be bad if one overlooked that more than 80% funding of the WHO comes from special interest groups. This rises major red flags. It is now almost three years into the pandemic, the scientific evidence is overwhelming in its wake with all politically introduced measurements and, in particular the new expression vaccines (mRNA and adenovector based vaccines) are causing more harm than they are beneficial. For your manuscript to have a lasting impact, my advice is to keep it purely scientific and this means purging it from any epidemiological numbers or calculations based on the mucosal COVID-19-tests. It becomes more and more clear that the SARS-COV-2 is not even as pathogenic as a simple H1N1 virus. E.g., when one compares the overestimated WHO death numbers putatively ascribed to COVID-19 pandemic with the Spanish flu, the Spanish flu death numbers were about 25-100 times higher in 1918/19. Comparatively, the age of COVID-19 deaths in Germany is on average about 83 years of age, which was only one year higher than the suspected average age at death in general. Of note, in 1918/19, it was also recorded that young adults were not spared. There is another naturally occurring beta-coronavirus, hCOV-OC43 (which also has a furin site) that is seen at the molecular level and is even more similar to flu-viruses with two independent pathogenicity surface markers, an esterase and the spike protein, different from SARS-COV-2 that has only the spike protein. The hCOV-OC43 is classified as cold virus. Why then is SARS-COV-2 not classified as cold virus? This classification should have happened even more so in the light of the SARS-COV-2 human infection trial in England (https://www.nature.com/articles/s41591-022-01780-9): Healthy probands were infected with a SARS-COV-2 variant and half of them did not even show symptoms. The other half showed only upper respiratory tract symptoms and had hardly any fever, both facts typical for a cold.

Therefore, I would state in the abstract and chiefly in the introduction of this manuscript that vaccines are the best functional defenses with the least financial burden against infectious viral diseases e.g. please see the vaccination success story against small pox. Moreover, the idea to develop a vaccine according to a country´s need backed by indigenous strains and phylogenetic analysis and strength of viral pathogenicity is a real good strategy. However, the authors need to indicate the mechanism that vaccination has in primarily preparing the adaptive immune responses, the humoral and cellular immune responses, to prevent cellular putative viral propagation. It should also become clear to the reader that initial adaptive immune responses as prepared by the vaccine beforehand to the future virus intruder need to cease for proper restoration of the individual´s health. This fine balance is normally not influenced by traditional vaccines. As the mRNA lingers more than 4 weeks in immunized individuals (Röltgen et al., 2022; https://www.cell.com/action/showPdf?pii=S0092-8674%2822%2900076-9), it is unclear whether this fine balance will be disturbed along with other interferences by the new mRNA vaccines. 

Another point to consider is the absolute risk reduction of the new expression vaccines: Why are the present, modern expression vaccines not more efficient than an absolute risk reduction of less than 2% (https://www.thelancet.com/action/showPdf?pii=S2666-5247%2821%2900069-0)?

The  vaccination developers face major challenges, partly by the virus through e.g. surface mutations/adaption and the compartmentalization in the human body e.g. into the lymph system, the brain, the intestine and the respiratory tract mucosal tissue, etc. And the respiratory tract immune defense is chiefly based upon the innate immune responses (IgA, gamma-delta-Tcells, NK-cells, innate lymphoid T-cells and pattern recognition receptors carrying cells). It is thought because of tolerance of the respiratory tract towards the many antigen interactions that, adaptive immune system responses have very limited access to the respiratory tract.

It is also important to note that T-cells once activated to one SARS-COV-2 variant will cross-react efficiently to any other SARS-COV-2 variant of concern (https://www.nature.com/articles/s41423-021-00767-9). Furthermore, T-helper cells also influence the maturation of B-cells. These observations indicate that, the surface changes of the virus are not important for escaping the vaccine activated adaptive immune responses. Thus humoral responses should be more used as a readout system of the effectivity to activate a meaningful adaptive immune response than thinking to be able to judge an in vivo effectiveness of the vaccine against relevant viral respiratory tract infect. It is not surprising that even after intensive scientific research for over 60 years there is no effective anti-RSV-Vaccine available.

For the proper analysis of vaccination efficiency in individuals the authors would need to perform randomized challenge double-blind infection trials, which definitively are out of the scope of the present work.

At the end of the introduction the authors might summarize the important data of the manuscript: the manuscript shows importance of silico driven variant designation in a traditional vaccination approach with whole inactivated virus…

For the Discussion:

Please interweave suggestion of the Introduction also in the Discussion and Conclusion. And extend it with following discussion points:

The authors need to point out why the traditional vaccine approach has advantages over the new expression vaccines: whole inactivated virus versus just the spike protein. 

Advantages of whole inactivated virus provides complex antigenicity to the adaptive immune response and resulting humoral responses have better cross-reactivity. 

Traditional vaccine strategies have been better explored and the risks are better defined whereas the risk/ benefit ratio of mRNA vaccines are not well defined and in the light of the recent publication in Vaccine (Serious adverse events of special interest following mRNA COVID-19 vaccination in randomized trials in adults; https://www.sciencedirect.com/science/article/pii/S0264410X22010283?via%3Dihub) the mRNA vaccines are too risky to be used. This publication was the logical outcome following previous publications that indicated that the concentration/ expression of the spike protein is uncontrollable (Röltgen et al., 2022, Cell), a toxicological nightmare, and that the artificial stabilized mRNA of the vaccine was found in the germinal centers of the secondary lymph organs (Röltgen et al., 2022, Cell), possibly disturbing the adaptive immune responses generally (Lederer et al., 2022, Cellhttps://pubmed.ncbi.nlm.nih.gov/35202565/).

If the authors´ work detailed on these suggestions they will have enough facts to amply contribute to scientific progress and worthy of being published in the journal, Vaccines.

Author Response

Sr. No.

 Comment

Reply

1

 As requested in your recent rebuttal, I will try to help with your introduction, discussion and conclusion.

If the authors´ work detailed on these suggestions they will have enough facts to amply contribute to scientific progress and worthy of being published in the journal, Vaccines.

Thank you for your valuable input. We appreciate your concerns not only for this study but also those that are published previously by others originating from different geographical regions. 

Also, per your suggestion, we have significantly improved the introduction, and discussion section of the manuscript (mentioned in track changes).

Dear reviewer,

Thank you for your valuable suggestions. We really appreciate your input to improve this manuscript.

If there is something that needs further to improve, please guide/suggest us and we will be happy to further revise and clarify the text accordingly.

Round 3

Reviewer 2 Report

The text needs EXTENSIVE REWRITING!...The English is too often poor and full or mistakes.

In several sentences, undue repetitions occur: l. 14; l. 73 for example

The name "rhesus macaques" needs an "s" when it is plural l. 29, 260, 264, 297, 299, 381;etc .

Table 1 requires more extensive explanations: What means the term "vaccine history" ? What is "CT"? How is it measured and expressed? What interest is there to list barcodes??

The labelling of the sera in Table 2 is different from that in Fig 3 and from that in Table 1!! How can the reader understand?

Why one does not find the abbreviations used in Table 4  in the long list of the Table of abbreviations at the end of the article?

Several sentences need to be rewritten! As an example, l. 367-370: "We interestingly found that the neutralisation titer at 28 days post-vaccination was 1:64 to 1:256 in all three animal models and reached 1:256 to 1: 1024 at 35 days, again in all the vaccinated animals (rabbits, mice and macaques)".

l. 279-280: " In comparison, serum samples from all the animals in control groups never showed any ability to neutralise the virus, as found by the observation of CPE".

l. 397: "for developing  inactivated vaccines in developing countries in the furure"

<Spelling mistakes, missing words or letters: l.41, 53, 57, 59, 60, 70, 134,179, 285, 311, 320 etc... Inappopriate use of "the": l. 249, 250, 279,...

Author Response

Reviewer 2

Sr. No.

 Comment

Reply

1

The text needs EXTENSIVE REWRITING!...The English is too often poor and full or mistakes.

Thank you for the suggestion. English has been improved in the revised manuscript by native English speaker.

2

In several sentences, undue repetitions occur: l. 14; l. 73 for example

Thank you for pointing out this mistake. These repetitions are corrected in the revised manuscript.

3

 The name "rhesus macaques" needs an "s" when it is plural l. 29, 260, 264, 297, 299, 381;etc .

Thank you for the suggestion. S has been added to Rhesus macaque in the revised manuscript.

4

Table 1 requires more extensive explanations: What means the term "vaccine history"? What is "CT"? How is it measured and expressed? What interest is there to list barcodes??

Vaccine history was basically the vaccination history of patients, either patients were vaccinated or not and which vaccine was administered.

CT is the cyclic threshold value which was measured by Real time PCR and is mentioned in the manuscript as well. As this is universal method, therefore the detail is not mentioned in the manuscript.

Barcodes were unnecessary, therefore have been removed in the revised manuscript.

5

The labelling of the sera in Table 2 is different from that in Fig 3 and from that in Table 1!! How can the reader understand?

Thank you for pointing out this mistake. We added accession numbers in the table 1 for better understanding. But Fig 3 is totally different as in that figure, we only showed the antibody titer of the vaccinated animals.

6

Why one does not find the abbreviations used in Table 4  in the long list of the Table of abbreviations at the end of the article?

Thank you for pointing out this mistake. Abbreviations used in Table 4 are also added in the revised manuscript.

7

Several sentences need to be rewritten! As an example, l. 367-370: "We interestingly found that the neutralisation titer at 28 days post-vaccination was 1:64 to 1:256 in all three animal models and reached 1:256 to 1: 1024 at 35 days, again in all the vaccinated animals (rabbits, mice and macaques)".

Thank you for the suggestion. These lines have been rephrased (l. 362-364) in the revised manuscript.

8

l. 279-280: " In comparison, serum samples from all the animals in control groups never showed any ability to neutralise the virus, as found by the observation of CPE".

Thank you for the suggestion. These lines have been rephrased (l. 278-280) in the revised manuscript.

9

l. 397: "for developing  inactivated vaccines in developing countries in the furure"

Thank you for the suggestion. These lines have been rephrased (l. 385-387) in the revised manuscript.

10

<Spelling mistakes, missing words or letters: l.41, 53, 57, 59, 60, 70, 134,179, 285, 311, 320 etc... Inappopriate use of "the": l. 249, 250, 279,...

Thank you for pointing out this mistake. Spelling mistakes have been corrected in the revised manuscript.

Dear Reviewer,

We tried our best to revise the manuscript for the improvement of English grammar for the readers.

We really appreciate your valuable suggestions to improve our manuscript. If there is something that needs further to improve, please guide/suggest us and we will be happy to further revise and clarify the text accordingly.

Reviewer 4 Report

There are minor changes that need to be addressed:

Lines 38-39: Please remove the sentence and the associated reference 2 “Till December 2022…globally (2).” These numbers as I indicated before are not factual.

Lines 39-41: The sentence “The index case…spread of the” is incomplete.

Line 49: please replace “nucleic acid based” with mRNA- and vector-based vaccines.

Lines 52-53: Please correct these lines to include: Regardless of the type, vaccines prepare the adaptive (humoral and cellular)…against the putative virus/bacterium/fungus and resulting immune responses prevent propagation of the pathogens in the individual.

Line 57:  please also include (Lederer et al., 2022, Cell; https://pubmed.ncbi.nlm.nih.gov/35202565/) with reference 7, which will be new “6” because reference “2” was removed.

Line 58 and following lines: I found “typos” throughout the manuscript, however, here, there is an accumulation of them, e.g. “8.., varriant, efeciently”…. Please, check for typos throughout the manuscript.

Line 65: It should be “viral respiratory tract infections”. The authors need to find another reference in this place supporting the idea that native immune responses are the main immune defense for viral respiratory tract infects.

As I previously commented IgA, gamma-delta-Tcells, NK-cells, innate lymphoid T-cells and pattern recognition receptors carrying cells are the mucosal defense.

Line 247: Can you please correct, “107.5 TCID50/ml” throughout the manuscript - it should read 107.5 virus particles. Please also include how you determined this value in the M&M.

Lines 315-318: Please rephrase this sentence as it is incomplete.

Author Response

Reviewer 4

Sr. No.

 Comment

Reply

1

 Lines 38-39: Please remove the sentence and the associated reference 2 “Till December 2022…globally (2).” These numbers as I indicated before are not factual.

Thank you for the suggestion. This line is removed in the revised manuscript

2

Lines 39-41: The sentence “The index case…spread of the” is incomplete.

Thank you for pointing out this mistake. The word disease was missing and now added in the revised manuscript.

3

Line 49: please replace “nucleic acid based” with mRNA- and vector-based vaccines.

Thank you for the suggestion. Nucleic acid based is replaced with mRNA in the revised manuscript.

4

Lines 52-53: Please correct these lines to include: Regardless of the type, vaccines prepare the adaptive (humoral and cellular)…against the putative virus/bacterium/fungus and resulting immune responses prevent propagation of the pathogens in the individual.

Thank you for the suggestion. These lines are rephrased as suggested in the revised manuscript.

5

Line 57:  please also include (Lederer et al., 2022, Cell; https://pubmed.ncbi.nlm.nih.gov/35202565/) with reference 7, which will be new “6” because reference “2” was removed.

Thank you for the suggestion. This reference has been added in the revised manuscript.

6

Line 58 and following lines: I found “typos” throughout the manuscript, however, here, there is an accumulation of them, e.g. “8.., varriant, efeciently”…. Please, check for typos throughout the manuscript.

Thank you for pointing out this mistake. We have checked the typos throughout the manuscript.

7

Line 65: It should be “viral respiratory tract infections”. The authors need to find another reference in this place supporting the idea that native immune responses are the main immune defense for viral respiratory tract infects.

As I previously commented IgA, gamma-delta-Tcells, NK-cells, innate lymphoid T-cells and pattern recognition receptors carrying cells are the mucosal defense.

Thank you for the suggestion. New reference has been added in the revised manuscript.

8

Line 247: Can you please correct, “107.5 TCID50/ml” throughout the manuscript - it should read 107.5 virus particles. Please also include how you determined this value in the M&M.

Thank you for the suggestion. 107.5 virus particles are written in the revised manuscript. It is also added in the virus titration heading in the M&M.

9

Lines 315-318: Please rephrase this sentence as it is incomplete.

Thank you for the suggestion. Line has been rephrased in the revised manuscript (l. 313-317).

Dear reviewer,

Thank you for your valuable suggestions. We really appreciate your input to improve this manuscript.

If there is something that needs further to improve, please guide/suggest us and we will be happy to further revise and clarify the text accordingly.
